# Pituitary Adenylate Cyclase Activating Polypeptide (PACAP) Reduces Oxidative and Mechanical Stress-Evoked Matrix Degradation in Chondrifying Cell Cultures

**DOI:** 10.3390/ijms20010168

**Published:** 2019-01-04

**Authors:** Eszter Szentléleky, Vince Szegeczki, Edina Karanyicz, Tibor Hajdú, Andrea Tamás, Gábor Tóth, Róza Zákány, Dóra Reglődi, Tamás Juhász

**Affiliations:** 1Department of Anatomy, Histology and Embryology, Faculty of Medicine, University of Debrecen, Nagyerdei krt. 98, H-4032 Debrecen, Hungary; szentleleky.eszter@dental.unideb.hu (E.S.); szegeczki.vince@anat.med.unideb.hu (V.S.); karanyicz.edina@anat.med.unideb.hu (E.K.); hajdu.tibor@anat.med.unideb.hu (T.H.); roza@anat.med.unideb.hu (R.Z.); 2Department of Anatomy, MTA-PTE PACAP Research Team, University of Pécs Medical School, Szigeti út 12, H-7624 Pécs, Hungary; andreatamassz@gmail.com (A.T.); dora.reglodi@aok.pte.hu (D.R.); 3Department of Medical Chemistry, University of Szeged, Faculty of Medicine, Dóm tér 8, H-6720 Szeged, Hungary; toth.gabor@med.u-szeged.hu

**Keywords:** matrix metalloproteinase, hyaluronidase, aggrecanase, mechanical stress, oxidative stress

## Abstract

Pituitary adenylate cyclase activating polypeptide (PACAP) is an endogenous neuropeptide also secreted by non-neural cells, including chondrocytes. PACAP signaling is involved in the regulation of chondrogenesis, but little is known about its connection to matrix turnover during cartilage formation and under cellular stress in developing cartilage. We found that the expression and activity of hyaluronidases (Hyals), matrix metalloproteinases (MMP), and aggrecanase were permanent during the course of chondrogenesis in primary chicken micromass cell cultures, although protein levels changed daily, along with moderate and relatively constant enzymatic activity. Next, we investigated whether PACAP influences matrix destructing enzyme activity during oxidative and mechanical stress in chondrogenic cells. Exogenous PACAP lowered Hyals and aggrecanase expression and activity during cellular stress. Expression and activation of the majority of cartilage matrix specific MMPs such as MMP1, MMP7, MMP8, and MMP13, were also decreased by PACAP addition upon oxidative and mechanical stress, while the activity of MMP9 seemed not to be influenced by the neuropeptide. These results suggest that application of PACAP can help to preserve the integrity of the newly synthetized cartilage matrix via signaling mechanisms, which ultimately inhibit the activity of matrix destroying enzymes under cellular stress. It implies the prospect that application of PACAP can ameliorate articular cartilage destruction in joint diseases.

## 1. Introduction

The structural and functional integrity of articular cartilage depends on the balance of appropriate extracellular matrix (ECM) secretion and elimination. The turnover rate of ECM macromolecules is very low in mature articular cartilage, and chondrocytes rarely divide, if at all. Moreover, hyaline cartilage is an avascular tissue without innervation. As a consequence of these natural features, this tissue has a limited regeneration in case of any damage. The unique internal architecture of articular cartilage is based on the well-organized orientation of collagen fibers comprising mostly type II collagen and supported by the high amount of hyaluronic acid and the major proteoglycan of cartilage, aggrecan [1]. Glycosaminoglycan side chains of proteoglycans and hyaluronic acid bind water, which ensures the high mechanical resistance of the tissue against pressure and shear stress [2]. In several disorders, such as osteoarthritis and rheumatoid arthritis, the composition of cartilage specific ECM can be partly destroyed with subsequent reduction of articular cartilage mechanical stability [1].

The expression of ECM components of hyaline cartilage is regulated by several signaling pathways. Kinases such as protein kinase A (PKA) [3], protein kinase C (PKC) [4] and mitogen-activated protein kinase (MAPK)-s [5] and phosphoprotein phosphatases (PP), such as PP2A [6] and PP2B [5,7] take part in the regulation of proper chondrogenesis and ECM production. A classic downstream target of PKA is a cyclic adenosine monophosphate (cAMP) response element-binding protein (CREB) transcription factor, which can regulate the expression of collagen type II [3]. Moreover, PKA regulates Sox9, which is the master transcription factor during cartilage formation activating the expression of aggrecan and collagen type II [5,8]. Disturbance of these signaling processes leads to an abnormal matrix production [5]. Generation of excessive amounts of reactive oxygen species during cartilage development, or the normal life cycle of cartilage results in reduced matrix production [5,7,9]. It is known that not only oxidative stress but also mechanical overload can lower the expression of collagens and aggrecan proteins [10]. In osteoarthritis, the release of pro-inflammatory cytokines and reduction of metabolic processes decrease the matrix production, while enhanced catabolism simultaneously destroys the normal architecture and matrix components of articular cartilage, triggering apoptotic and necrotic processes [11].

During these diseases, the demolition of ECM is the result of the activity of various matrix degrading enzymes [12]. One of the most important groups of these enzyme families is the matrix metalloproteinases (MMP). These enzymes, such as MMP1, 7, 8, 9, and 13, disintegrate and collagen-specifically digest the matrix components [13]. MMP1, 8, and 13 are considered as the major collagenases that are able to disintegrate the triple helix of collagens. MMP7 is a matrilin specific proteolytic enzyme and MMP9 is a gelatinase that is responsible for collagen type I and type X destruction [12,13]. Other families of the matrix degrading enzymes can destroy the cartilage specific proteoglycans, ADAMTS4 and ADAMTS5 are described as major aggrecanases of osteoarthritis [14]. Hyaluronidases degrade hyaluronic acid, which ultimately leadsto the disintegration of the aggrecan–hyaluronan network in the cartilage matrix [15]. Four cartilage-specific hyaluronidases have been identified in the last decade [2].

Pituitary adenylate cyclase activating polypeptide (PACAP) is an evolutionary well-conserved neuropeptide, first described in the central nervous system [16]. Its isolation detailed function has been described not only in the central nervous system, but in various peripheral organs [17]. The neuropeptide plays an important role in the development of gonads [18], teeth [19], bone [20], and cartilage [21]. Furthermore, it can prevent harmful cellular effects of ischemic conditions, oxidative stress, and inflammation [7,22,23,24]. PACAP can act on three main G protein coupled receptors, namely, PAC1, VPAC1, and VPAC2, from which PAC1 has the highest affinity to the neuropeptide while the other two receptors bind PACAP and VIP equally [17,25]. PACAP binding to its receptors triggers the activation of adenylate cyclase, and subsequently the increase of intracellular cAMP, which induces PKA activity [17]. PKA can phosphorylate CREB and Sox transcription factors, considered as major regulators of chondrogenesis [7]. PACAP has been shown to have several signaling crosstalks with bone morphogenetic protein (BMP), wingless int1 (WNT), or hedgehog (HH) pathways, which are also important in osteo- or chondrogenesis [21]. PACAP has been demonstrated to play an important role in inhibition of matrix degradation in osteoarthritis or in rheumatoid arthritis [26]. It is known to have a positive effect on hyaluronan synthesis or collagen type II secretion; moreover, it increases the expression of Sox9 and elevates phospho-Sox9 in oxidative stress and mechanical load in chondrogenic cell cultures [7,10]. However, only sporadic data exist concerning its role in the regulation of ECM degrading enzymes.

In this study, we present data on the expression and activity of various matrix degrading enzymes during the course of in vitro chondrogenesis and demonstrate that PACAP signaling attenuates the activation of matrix metalloproteinases, hyaluronidases, and aggrecanase during oxidative and mechanical stress in chondrogenic cell cultures.

## 2. Results

### 2.1. Hyaluronidase Expression in Chondrifying Cells

Hyaluronidases may have a function during chondrogenic differentiation or in the remodeling of cartilage ECM. We detected the presence of mRNAs of all hyaluronidases by reverse transcription followed by polymerase chain reactions (RT-PCR) in cells of micromass cell cultures. The mRNA expression of *Hyal1* and *Hyal3* was most prominent on days 2 and 3, while *Hyal 4* expression peaked on day 4. All Hyal genes became lowered by the end of culturing, when mature chondrocytes dominate the cell population (Figure 1A). Interestingly, only Hyal2 protein expression was similar to the mRNA expression, showing a decreased expression by the end of culturing, while the other hyaluronidase enzyme proteins were detected at variable levels (Figure 1B). We also measured the activity of these enzymes and increased activity was detected on day 2 and 3 of culturing on days of cartilage specific matrix formation (Figure 1C).

### 2.2. Identification of MMPs in Chondrifying Cell Cultures

After the expression of hyaluronidases, we examined the possible matrix metalloproteinases playing a role in chondrogenesis. mRNA expression of *MMP1* reduced until day 3 of culturing and then elevated until the end of culturing period (Figure 2A). *MMP7* and *MMP13* mRNA expression increased continuously until day 6 of chondrogenesis (Figure 2A). The mRNA expression of *MMP8* became strong on the days of chondrogenic transformation, then decreased by the end of culturing (Figure 2A). *MMP9* mRNA diminished from days 2 and 3 of culturing and was barely detectable on the last day of culturing (Figure 2A). Protein expression of MMP1 was not correlated with the mRNA expression as it was elevated on day 3 of differentiation, then reduced to the end of culturing. The protein level of MMP7 decreased on days of differentiation and then elevated back to control level to day 6 of chondrogenesis (Figure 2B). MMP8 showed a peak-like pattern on days of chondrogenic differentiation (Figure 2B). Expression of MMP9 was hardly detected except for on day 2 of culturing when a strong signal appeared (Figure 2B). MMP13 continuously increased until the end of the culturing period (Figure 2B).

As the protein expression pattern of these enzymes was not always correlated with their catalytic activity, we investigated their activity with zymography using different substrates. Collagen type I substrate shows the activity of MMP9 at 78 kDa, which elevated on day 1 and 2 of chondrogenic differentiation. MMP13 also cleaves collagen type I at 54 kDa and its activity increased on days 2 and 3 of culturing considered days of chondrogenic differentiation (Figure 2C). Activity of MMP9 was further investigated using gelatin substrate, which visualizes its active form of MMP9 at 76 kDa. Similar to MMP13, the active MMP9 showed a peak on days 2 and 3 of chondrogenic differentiation (Figure 2C). Cleavage of casein is specific for MMP1 activity, which continuously reduced by the end of culturing (Figure 2C).

### 2.3. Aggrecanase Activity in Chondrogenesis

Proper extracellular matrix production is also based on the precise timing of aggrecan expression and its well-regulated turnover. mRNA expression of *ADAMTS4*, a specific proteolytic enzyme of aggrecane, showed a peak on days 2 and 3 of culturing (Figure 3A). Its protein expression continuously reduced until the end of chondrogenesis (Figure 3B). On the contrary, the activity of ADAMTS4 remained constant during chondrogenic differentiation (Figure 3C).

### 2.4. Effects of PACAP and Various Stress on Metachromatic Cartilage Formation

In line with our previously reported data [7], addition of PACAP on days 2 and 3 of culturing increased the cartilage production to 123% (Figure 4), while 4 mM H_2_O_2_ reduced [5] the metachromatic matrix production to 22% (Figure 4). Mechanical stimulus also increased matrix production [10], while using an increased intensity of mechanical load elevated the cartilage formation to 125% (Figure 4). Administration of PACAP was able to compensate the harmful effect of H_2_O_2_ and increased the cartilage formation to 85% (Figure 4). Presence of PACAP in the increased mechanical stress resulted in an elevated matrix formation to 127% (Figure 4). Interestingly, the application of mechanical stress with H_2_O_2_ administration diminished matrix production to 25% (Figure 4). Application of the two stress factors with PACAP administration reduced cartilage formation to 85% (Figure 4). The applied stress factors did not induce significant apoptosis or necrosis (Appendix A).

### 2.5. Various Cellular Stress Increase Hyluronidase Activity

Administration of H_2_O_2_ increased the mRNA expression of *Hyal1*, *Hyal2*, *Hyal3*, and *Hyal4* (Figure 5A), and a similar effect was detected after mechanical load (Figure 5A) and during the combination of the two stress factors (Figure 5A). The presence of PACAP reduced the mRNA expression of *Hyal1* and *Hyal4*, but interestingly elevated the mRNA expression of *Hyal2* and *Hyal3* (Figure 5A). Administration of PACAP with H_2_O_2_ and/or mechanical stress also reduced the mRNA expression of *Hyal1*, *Hyal3*, and *Hyal4* (Figure 5A). *Hyal2* showed a similar expression pattern as it was detected during single stress factors. Investigation of protein expression did not always correlate with the mRNA expression. H_2_O_2_ administration increased the protein expression of all hyaluronidases (Figure 5B). Mechanical stress resulted in an elevated protein expression of Hyal1, Hyal3, and Hyal4 (Figure 5B). On the contrary, the protein level of Hyal2 was hardly detectable after mechanical load (Figure 5B). The combined application of the two stresses elevated the expression of Hyal1, Hyal3, and Hyal4. Administration of PACAP neuropeptide together with H_2_O_2_ reduced the protein expression of Hyal1, Hyal2, and Hyal4, but further elevation was detected in Hyal3 (Figure 5B). PACAP did not completely prevent the harmful effect of mechanical stress and did not significantly lower the protein expression of Hyal1 and Hyal4 (Figure 5B). The protein level of Hyal2 and Hyal3 was elevated by mechanical load (Figure 5B). Moreover, the protein expression of Hyal1 and Hyal4 decreased in the presence of PACAP and stress factors, but protein signals of Hyal3 became stronger (Figure 5B). No significant changes were visible in Hyal2 (Figure 5B). Summarized activity of hyaluronidases elevated in the presence of H_2_O_2_ and/or mechanical stimulus, moreover, combined stress led to a further increase (Figure 5C). PACAP was not able to reduce the effect of H_2_O_2_, but decreased the hyaluronidase activity during mechanical stress. Interestingly, the hyaluronidase activation evoked by combined cellular stress was reduced by PACAP (Figure 5C).

### 2.6. Expression of MMPs Affected by PACAP Treatment

The treatment of chondrogenic cultures with H_2_O_2_ increased the *MMP1* and *MMP7* mRNA expression, but no alterations were found in the expression of *MMP8*, *MMP9*, and *MMP13* (Figure 6A). Similarly, mRNA expression of *MMP1*, *MPP7*, and *MMP9* elevated after mechanical load and no alterations were detected in *MMP8* and *MMP13* expression (Figure 6A). *MMP1* and *MMP7* mRNA expression was reduced by PACAP treatment, while no significant changes were detected in the mRNA expressions of other MMPs (Figure 6A). The presence of PACAP normalized the expression of *MMP1* and *MMP7* during H_2_O_2_ treatment or mechanical load (Figure 6A). However, no significant alteration was detected in the *MMP8* and *MMP13* mRNA expression after the same stress (Figure 6A). mRNA expression of *MMP9* increased in the presence of PACAP and H_2_O_2_ or mechanical load (Figure 6A). Similar to the mRNA profile, the protein expression of MMP1 and MMP7 increased in the presence of H_2_O_2_ and after mechanical load or in the result of the combined application of the two stress (Figure 6B). In case of MMP8 and MMP13, the protein expression changed differently than the mRNA expression; protein levels increased after the various stress, while MMP9 protein responded similarly to mRNA expression (Figure 6B). The presence of the neuropeptide did not significantly alter the expression of the MMPs, but reduced their expression to the control level after oxidative stress (Figure 6B). Application of PACAP during mechanical stress diminished the expression of MMP1, MMP7, and MMP13 and significantly reduced MMP9 expression, but increased the protein level of MMP8 (Figure 6B). The combined application of mechanical load and H_2_O_2_ with PACAP reduced the expression of MMP1, MMP7, MMP8, and MMP13, but not the expression of MMP9 (Figure 6B). In zymography, collagen type I was used to detect the activity of MMP9 and MMP13 at 78 and 54 kDa. Oxidative stress doubled the activity of MMP9 and resulted in a six-fold increase of MMP13. Furthermore, the strong mechanical load, similar to the oxidative stress, elevated the activity of MMP13 to six-fold (Figure 6C). Interestingly, the combination of the two harmful effects was not additive and resulted only in a two-fold elevation (Figure 6C). Surprisingly, PACAP treatment elevated the activity of MMP13 to two-fold, but the neuropeptide was able to reduce MMP13 activity during oxidative stress and/or mechanical load (Figure 6C). PACAP did not decrease the collagen type I cleavage capability of MMP9 in oxidative stress, but it reduced the activity of this enzyme during mechanical load (Figure 6C). For better understanding, the function of MMP9 gelatin was also used as a substrate, where the precursor and the active forms of MMP9 can be distinguished. The active form of MMP9 was significantly higher in H_2_O_2_ treatment and PACAP was able to reduce it to control level (Figure 6C). Mechanical load did not significantly alter the amount of active MMP9 and PACAP had no effect on MMP9 activity (Figure 6C). We mixed the gel with casein as well, which can be a substrate of MMP1 at 54 kDa. Both oxidative stress and mechanical load augmented the MMP1 activity (Figure 6C). PACAP did not alter the casein cleavage ability of MMP1, but reduced its activity during oxidative stress and/or strong mechanical load (Figure 6C).

### 2.7. Aggrecanase Function during Oxidative and Mechanical Stress

For the detection of aggrecan cleavage, we monitored the expression of ADAMTS4. mRNA expression of this enzyme responded in an opposite manner to the two different stressors—H_2_O_2_ elevated it, while mechanical load diminished it (Figure 7A). Application of PACAP was able to lower the mRNA level of *ADAMTS4* and normalized its expression during oxidative stress, while it remained barely detectable when mechanical load and PACAP were applied simultaneously (Figure 7A). Protein expression of ADAMTS4 was in correlation with the mRNA expression and was elevated in the presence of H_2_O_2_, but this increase was reduced by PACAP treatment. Surprisingly, PACAP as well as mechanical load resulted in a double fold elevation in the expression of ADAMTS protein. Furthermore, effects of mechanical load were not compensated by PACAP addition (Figure 7B). Activity of aggrecanases was increased by addition of H_2_O_2_ increasing after one hour of oxidative stress (Figure 7C). Moreover, the highest activity was measured during the combined application of the two stresses (Figure 7C). PACAP and mechanical load, similar to the protein expression, resulted in an elevation of ADAMTS with aggrecanase activity (Figure 7C). PACAP reduced the aggrecanase activity to control the level in the presence of mechanical loading and after the combined application of the two stresses. On the other hand, ADAMTS activity responded with a moderate decrease only to the PACAP treatment when oxidative stress was applied without mechanical stress (Figure 7C).

## 3. Discussion

Differentiation in high density chondrogenic cell cultures is a complex process with precisely regulated intracellular and extracellular events leading to proper cartilage ECM formation. Initially, elongated chondroprogenitor cells start to proliferate rapidly; form cell aggregations; and degrade hyaluronan, type I collagen, and fibronectin rich mesenchymal-like ECM [6]. Next, the differentiated cells become round and synthesis of cartilage specific ECM molecules, such as aggrecan and collagen type II, begins. During the early steps of cartilage formation, establishment of a cartilage specific ECM and degradation of the immature mesenchymal ECM occur simultaneously [27]. This dynamic change and turnover of the ECM requires the presence of several matrix degrading enzymes such as matrix metalloproteinases [28], hyaluronidases [15], and aggrecanases [29]. As only sporadic data are available in relation to the expression and activity of these enzymes during chondrogenesis, we first aimed to explore this field. In chicken limb bud-derived primary chondrogenic cell cultures, chondroprogenitor cells differentiate to chondroblasts in six days, recapitulating cellular and ECM changes of the in vivo limb-cartilage formation. We demonstrated the presence of Hyal1, 2, 3, and 4 mRNA and protein expression in these cultures, but their mRNA and protein expression was not synchronized, suggesting that intercellular signals induce the translational activation of *Hyal* mRNAs in a paracrine regulation. In chondrogenesis, Hyals can be activated and digest the pericellular hyaluronan coat, through which cell metabolism is regulated. Hyals can also cleave bigger size hyaluronic acid to alter the organization and stability of aggrecan. This idea is supported by the observation that Hyal2 deficient mice show cartilage development disorders and osteoarthritic like cartilage degeneration [30]. We demonstrated that Hyal2 and Hyal3 protein expression reduced on days of chondrogenic differentiation, while Hyal4 was elevated and Hyal1 was not altered on these days. These findings indicate that Hyal2 and Hyal3 play a role in the regulation of proper hyaluronan content of cartilage ECM. A weak expression of Hyal4 was shown in dermal fibroblasts, but no Hyal4 was detected when cells were cultured under chondrogenic conditions [31]. It has been shown that Hyal4 does not have hyaluronidase activity, but is tissue specifically present in muscle tissue, testis, or placenta [32]. Our data show that activity of hyaluronidases was increased on days of chondrogenic differentiation, suggesting the importance of a dynamic change of extracellular hyaluronan content during commitment of chondroprogenitor cells.

Protein components of ECM are degraded mostly by matrix metalloproteinases (MMPs). The MMP family can be divided into various groups according to their substrate specificity, such as collagenases, gelatinases, or matrilysins [13]. A particular collagenase family including MMP1, MMP8, and MMP13 cleaves collagen type I, II, and X [13]. These MMPs all are present in chondrogenic high density cell cultures. MMP8 had the highest protein expression on days of differentiation, as well as with increased MMP13 activity. As MMP13 plays a role in terminal differentiation of chondrocytes [33], this observation suggests a function in collagen turnover during cartilage matrix production. Interestingly, MMP1 protein expression and activity continuously decreased in the differentiation process, reflecting on a possible role of this enzyme in the removal of the immature ECM in order to prepare for the replacement by cartilage specific ECM molecules. As the active form of MMP9, which can break down the denatured form of collagen type I, gelatin [13], was elevated on days of 2 and 3, showing that the matrix of the differentiating tissue turns into a cartilage specific matrix. MMP9 and MMP13 have been shown to be regulated by PTHrP and determine the matrix degradation of chondrocytes [34] and take part in normal cartilage morphology. MMP7 is part of the matrilysin family having gelatinase activity, and is frequently expressed in healthy tissues [13]. Its expression decreased in chondrogenic transformation, which shows the involvement of proper cartilage matrix production. In hMSC cultures, MMP7 is shown to have crosslinks with chondroitin sulphate binding peptides [35], which further strengthen its role in appropriate chondrogenesis. Our observations support the idea that a complex network of MMPs with a temporarily precisely regulated activity pattern takes part in the transformation of ECM during chondrogenesis. We also demonstrated that ADAMTS4, a member of the aggrecanase family, was also present in in vitro cartilage formation with a constant protein expression and low activity. ADAMTS4 can be regulated by NFAT or Runx2 and has a function in physiological and pathological aggrecan degradation [36], indicating that dynamic production and remodeling of the tissue needs a fine tuned balance between matrix production and transformation.

PACAP is a short, evolutionary well-conserved neuropeptide thathas anti-apoptotic effects and protects tissues from harmful influences such as ischemic conditions, oxidative stress, or inflammations [21]. PACAP signaling, through the activation of its main receptor PAC1, is connected to several other fundamental pathways, such as WNT, hedgehog, or BMP [10,21]. PACAP is secreted by chondrogenic cells of high density cultures (HDC) and these cells express its receptors [7], moreover, PAC1 receptor is also shown in osteoarthritic cartilage [26]. The addition of PACAP into the medium of HDCs reduced the harmful effects of oxidative stress [7] and normalized the cartilage specific matrix production [10]. We have also demonstrated that PACAP had a positive effect on hyaluronan synthesis and aggrecan production of cells in HD cultures [7]. PACAP positively regulates Sox9 activation and exogenous PACAP prevented the Sox9 reducing effect of H_2_O_2_ [7]. It was also shown to exert an anti-inflammatory function in osteoarthritis [26]. Although positive effects of PACAP have been proven on matrix synthesis, only sporadic data exist about receptor activation in matrix degradation [37].

Degeneration of articular cartilage can be triggered by oxidative stress or mechanical overload, both of which can induce inflammation and lead to development of osteoarthritis. Activation of several proteolytic enzymes is enhanced upon these stressors and leads to destruction of the cartilage specific ECM. Ultimately, the disintegration of ECM demolishes the weight baring capacity of the tissue [38]. Previously, we demonstrated that oxidative stress reduced the metachromatic cartilage matrix [5,9], while optimized cyclic mechanical load increased matrix production in high density cultures [39]. Nonetheless, an increased expression of collagen type X was also demonstrated in HDCs after mechanical load [10]. In the present study, we applied a mechanical stimulation that mimicked the overload.

To demonstrate the effects of PACAP on matrix degrading enzymes, we used high density cell cultures as an in vitro model where the chondrogenic differentiation occurs on days 2 and 3 of the six-day-long culturing. Mechanical overload has been shown to induce the activation of Hyal1 and 2 and triggered inflammatory processes in mature articular chondrocytes [40], but no data are available concerning the effect of oxidative stress on Hyal synthesis and activity during chondrogenesis. The applied mechanical overload and oxidative stress did not result in significant apoptotic or necrotic processes (Appendix A). We demonstrated that oxidative stress increased the expression and activity of Hyal1, 2, 3, and 4, but PACAP did not have any positive or negative effects on hyaluronidases function. On the contrary, mechanical stress induced activation of hyaluronidases, which was reduced by PACAP addition. It has been reported that appropriate mechanical load increased the expression of hyaluronic acid [7] and PACAP addition also increased the presence of the glucosaminoglycan [7,10]. However, their effects were not additive. On the contrary, H_2_O_2_ reduced hyaluronan content of developing cartilage, which was prevented by PACAP treatment [7]. It has been published that cyclic tensile load induces the activation of Hyals and can regulate hyaluronan catabolism in synovial fluid [41]. These findings suggest that PACAP has an important function in hyaluronic acid turnover via the activation of hyaluronic acid synthases and regulation of hyaluronan degradation. However, the addition of hyaluronic acid is able to prevent the harmful effects of oxidative stress [42] and induces migration of tumor cells [43], but no clear data can be found in connection with hyaluronan catabolism.

Matrix metalloproteinases can degrade the collagen content of cartilage, and various cellular stress such as mechanical overload, inflammatory effects, and reactive oxygen species can activate them. Mechanical force can lead to the changing of cellular response via triggering mechanotransductional pathways [44]. Mechanical overload can regulate the expression and activity of matrix metalloproteinases. Increased mechanical stress in rheumatoid arthritis results in a rapidly developing inflammation and elevated MMP1 and MMP13 expression, which can induce the degradation of cartilage and subchondral bone matrix [44]. Osteoarthritis can induce inflammation and elevated MMP expression [44]. In increased mechanical stress, the activity of MMP1, 7, 8, 9, and 13 is elevated, which results in the degradation of cartilage matrix [44]. In our experiments, mechanical overload induced higher protein expression of all investigated MMPs and elevated the activity of MMP1 and MPP13. The addition of PACAP reduced the activation of these metalloproteinases. Similar results were shown in murine cartilage as the increased mechanical force induced the activity of MMP13 [45], and PKA modulation has an effect on MMP13 secretion [46]. These results also suggest that the PACAP activated PKA signaling pathway can inhibit MMP13 expression and activation, which helps to protect the appropriate matrix composition of cartilage. Moreover, it has been shown that cAMP–PKA signalization inhibits MMP1 activation [47], which implies PACAP’s importance in the normalization of MMPs activation. MMP8 function can be dual as it has an important role in osteogenic differentiation by degradation of the chondrogenic matrix [27], and also has a protective role in arthritis formation [48]. Our results suggest that inhibition of MMP8 protein expression by PACAP in oxidative stress normalizes and keeps in balance the cartilage matrix production. Generally, elevation of cAMP concentration inhibits the expression of MMPs [49], which further strengthen the idea that PACAP induced adenylate cyclase activation can control the function of MMPs. On the contrary, active MMP9 is not involved in this regulation as only slight alterations were detected in mechanical load. Connection of MMP9 and PKA pathway is contradictory and cell-type specific as in monocytes, the activation of MMP9 can occur independently of PACAP via p38 activation [37], but PACAP can directly activate MMP9 in lymphocytes and macrophages [50]. We found that oxidative stress also increased the protein expression and the activation of MMP9, which was normalized by PACAP addition. As the mRNA expression of *MMP9* remained unaffected in oxidative stress, either with or without PACAP addition, while protein expression changed in both cases, it suggests that PACAP influences MMP9 activity without modulation of its gene expression. Indeed, the canonical downstream kinase of PACAP, PKA, can regulate Runx2 activation [20], which in turn can induce the expression of MMP9 [51]. Runx2 regulates the terminal differentiation of chondrocytes [52], inducing chondrocyte hypertrophy and matrix calcification. Runx2 has a direct connection with IHH. As the hedgehog signaling pathway is inhibited by PACAP signaling [10,53], it suggests that Runx2 activation can occur in PACAP dependent or independent ways. We can hypothesize that PACAP addition during oxidative stress can normalize PKA activity, which fine tunes the activation of Runx2-dependent MMP9 expression, maintaining the cartilage specific matrix production and that this pathway is in crosstalk with hedgehog signaling.

A similar effect was observed in the case of MMP7, as elevated expression was detected in oxidative and mechanical stress, which was diminished in the presence of PACAP. MMP7 was shown to have a function in pericellular matrix degradation [54] and it has a direct connection with protein kinase activation [55]. In chondrosarcoma cells, it has been demonstrated that increased shear stress via cAMP elevation induced MMP7 expression [56]. Our data suggest that normalization of the cAMP dependent pathway via PACAP addition can inhibit MMP7 activation too, through which PACAP protects abnormal matrix degradation. MMP7 also has aggrecane cleavage capability [57], suggesting that PAC1 receptor activation has a direct connection with aggrecanase function.

We detected elevated aggrecanase activity in oxidative and mechanical stress. Moreover, when PACAP was added without application of cellular stress, it also increased the expression of ADAMTS4 and the activity of aggrecenases. On the contrary, PACAP diminished the expression and activity of aggrecenase to control level during oxidative and mechanical stress. ADAMTS4 expression and activation is increased in osteoarthritis and rheumatoid arthritis [58]. Furthermore, ADAMTS4 is suppressed by Sox9 transcription factor activation [59] and MMP-induced activation of aggrecanase has been shown in the intervertebral disc [60]. Additionally, nitric oxide triggers the activation of aggrecanases in meniscal cells [61]. Mechanical load activates cAMP production and ADAMTS4 expression time dependently [62], suggesting a communication between the two processes. The presence of neuropeptides such as vasoactive intestinal polypeptide (VIP) inhibits the expression of ADAMTS4 [63], but no data can be found in connection with PACAP. One possible link can be Runx2 and ß-catenin, which are in connection with ADAMTS4 activation [63]. Both of these pathways are in connection with PAC1 receptor activation and PKA signaling [64]. Altogether, aggrecanase activation can be regulated in two signaling processes, partly by the increased Sox9 function in PACAP presence, which is able to reduce ADAMTS4 expression [7], and a PACAP dependent crosslink via the Runx2 activation [63] also regulates ADAMTS4 activation.

In conclusion, PACAP can be a potent substance that can positively regulate matrix production in articular cartilage, particularly in the presence of various cellular stress conditions, such as mechanical overload or oxidative stress, important in the progression of matrix degradation in osteoarthritis or rheumatoid arthritis.

## 4. Materials and Methods

### 4.1. Cell Culturing

Chicken embryos of Hamburger–Hamilton stage 22–24 were used for establishment of primary chondrogenic cell cultures. Briefly, distal parts of chicken limb buds were removed and after washing in calcium-magnesium free (CMF)–phosphate buffered saline (PBS), they were digested in trypsin solution for 1 h. Cells were harvested and washed in foetal bovine serum (FBS) and a 1.5 × 10^6^ cells/mL concentration was set. We inoculated cell suspension droplets on the bottom of six-well plates (Eppendorf, Hamburg, Germany). The day of inoculation is considered as day zero. Colonies were nourished with Dulbecco’s modified eagle medium (Sigma-Aldrich, St. Louis, MO, USA), supplemented with 10% foetal calf serum (Gibco, Gaithersburg, MD, USA) and were kept at 37 °C in the presence of 5% CO_2_ and 95% humidity in a CO_2_ incubator. The medium was changed every second day. Differentiation of chondrogenic cells parallel with the increased cartilage specific ECM production peaked on the second and third days of culturing. On day 6, mature chondrocytes surrounded by a cartilage specific matrix, containing a high amount of hyaluronic acid, collagen type II, and aggrecan, could be detected.

### 4.2. Application of PACAP, H_2_O_2_, and Mechanical Stress

PACAP 1-38 at 100 nM (stock solution: 100 μM, dissolved in sterile distilled water) was used as an agonist of the PAC1 receptor. PACAP application was continuous from day 1. In order to generate oxidative stress, 4 mM H_2_O_2_ was administrated to the culture medium on the second and third days of culturing for 2 × 20 min. Mechanical stress (MS) was also applied on the second and third days for 30 min. A custom-made instrument described in Juhász et al. [39] was applied. Mechanical load was transmitted to the cell cultures by pedicles dipping in the culture medium four times per minute. These pedicles drop 2 mm deep into the medium with approximately 600 Pa force without touching the cells themselves. This movement generates hydrostatic pressure and fluid shear [39]. Timing was set to induce overloading the developing cells. In a combined application setup, cultures were exposed to both H_2_O_2_ and mechanical stress and at the presence of PACAP, cultures were exposed to H_2_O_2_ and MS treatment, or all treatments were applied on chondrogenic cells. Control cultures were kept in identical culture conditions without any treatments.

### 4.3. Light Microscopical Analysis

Droplets of limb bud mesenchymal cells of different experimental groups in sizes of 100 μL were cultured on the surface of 3 cm diameter round coverglass placed into the wells of six-well culture plates. High density cultures were established and on day 6, they were fixed in a 4:1 mixture of absolute ethanol and 40% formaldehyde. After fixation, 0.1% dimethylmethylene blue (DMMB) dissolved in 3% acetic acid was used to stain the cultures. To remove the dye, excess cultures were washed in 3% acetic acid then mounted in DPX (Sigma-Aldrich). Photomicrographs were taken using an Olympus DP72 camera on a Nikon Eclipse E800 microscope (Nikon Corporation, Tokyo, Japan). For semiquantitaive detection of the amount of metachromatic matrix, we dissolved back toluidine blue (TB; pH 2; Reanal, Budapest, Hungary) from cell cultures on day 6. This method provides a good approximation of the amount of formed cartilage, as was described in Matta et al. [65]. DMMB and TB metachromatic staining procedures were carried out on separate colonies from the same experiments; DMMB-stained specimens are only shown as visual representations of TB assays. The absorbance of the metachromatic cartilaginous areas was measured in three cultures of each experimental group in three independent experiments.

### 4.4. RT-PCR Reactions

Cell cultures were dissolved in Trizol (Applied Biosystems, Foster City, CA, USA), 20% RNase free chloroform was added, and the samples were centrifuged at 4 °C at 10,000 rpm for 15 min. Samples were incubated in 500 µl of RNase free izopropanol in −20 °C for 1 h, then total RNA was harvested in RNase free water and stored at −20 °C. The assay mixture for reverse transcriptase reaction contained 2 µg RNA, 0.112 µM oligo(dT), 0.5 mM deoxynucleotide triphosphate (dNTP), 200 units high capacity RT (Applied Bio-Systems, Foster City, CA, USA) in 1× RT buffer. For the sequences of primer pairs and further details of polymerase chain reactions, see Table 1. Amplifications were performed in a thermal cycler (Labnet MultiGene™ 96-well Gradient Thermal Cycler; Labnet International, Edison, NJ, USA) in a final volume of 11 μL (containing 0.5 μL forward and reverse primers (0.4 μM), 0.25 μL dNTP (200 μM), and five units of Promega GoTaq^®^ DNA polymerase in 1× reaction buffer) as follows: 95 °C for 2 min, followed by 35 cycles (denaturation, 94 °C, 1 min; annealing at optimized temperatures as given in Table 1 for 1 min; extension, 72 °C, 90 s), and then 72 °C for 10 min. PCR products were analyzed by electrophoresis in 1.2% agarose gel containing ethidium bromide. GAPDH was used as internal control. Optical density of signals was measured using ImageJ 1.40 g freeware (http://rsbweb.nih.gov/ij/) and results were normalized to the optical density of day 0 or untreated control cultures.

### 4.5. Western Blot

Three-day-old cell cultures were washed in physiological NaCl solution and harvested. After centrifugation, cell pellets were suspended in 100 μL of homogenization RIPA (radio immuno precipitation assay)-buffer (150 mM sodium chloride; 1.0% NP_4_0, 0.5% sodium deoxycholate; 50 mM Tris, pH 8.0) containing protease inhibitors (Aprotinin (10 μg/mL), 5 mM Benzamidine, Leupeptin (10 μg/mL), Trypsine inhibitor (10 μg/mL), 1 mM PMSF, 5 mM ethylene diamine tetra-acetic acid (EDTA), 1 mM ethylene glycol-bis(β-aminoethyl ether)-*N*,*N*,*N*′,*N*′-tetra acetic acid (EGTA), and 8 mM Na-Fluoride, 1 mM Na-orthovanadate). Samples were stored at −70 °C. Suspensions were sonicated by pulsing burst for 30 s at 40 A (Cole-Parmer, Vernon Hills, IL, USA). For Western blotting, total cell lysates were used. Samples for SDS-PAGE were prepared by the addition of Laemmli electrophoresis sample buffer (4% SDS, 10% 2-mercaptoethanol, 20% glycerol, 0.004% bromophenol blue, 0.125 M TrisHCl pH 6.8) to cell lysates to set equal protein concentrations of samples, and boiled for 10 min. About 10 µg of protein was separated by 7.5% SDS-PAGE gel for detection. Proteins were transferred electrophoretically to nitrocellulose membranes. After blocking with 5% non-fat dry milk in phosphate buffered saline with 0.1% Tween 20(PBST), membranes were washed and exposed to the primary antibodies overnight at 4 °C in the dilution as given in Table 2. After washing for 30 min in PBST, membranes were incubated with anti-rabbit IgG (Bio-Rad Laboratories, Hercules, CA, USA) in 1:1500 or anti-mouse IgG (Bio-Rad Laboratories, Hercules, CA, USA) in 1:1500 dilution. Signals were detected by enhanced chemiluminescence (Advansta, San Jose, CA, USA) according to the instructions of the manufacturer. Signals were developed with gel documentary system (Fluorchem E, ProteinSimple, San Jose, CA, USA). The optical density of Western blot signals was measured using ImageJ 1.40 g freeware and results were normalized to that of untreated control or day zero cultures.

### 4.6. Hyaluronidase Assay

The medium of cell cultures was harvested from control conditions directly after treatments, as well as 30 min after the stress. Harvested mediums were kept in −70 °C. For the hyaluronidase assay, we used a turbidimetric assay as follows. Reagent A (300 mM sodium-phosphate, pH 5.35, (Sigma-Aldrich)), reagent B (hyaluronic acid, pH 5.35 (Sigma-Aldrich)),reagent C (Hyaluronic acid, (Sigma-Aldrich)), reagent D (bovine serum albumin (BSA) and NaCl and NaP (Sigma-Aldrich)), reagent E (acidic albumin solution (Sigma-Aldrich)), and reagent G (hyaluronidase enzyme solution (Sigma-Aldrich)) were made. From reagent G, D, and C, we made a standard row, and from samples and reagent D and C, working solutions were established. Supernatants were incubated at 37 °C on 80 rpm for 45 min, after which 500 µL from all samples and 2.5 mL of reagent E were added to working solutions. Three times, 100 µL of supernatants were separated and the absorbance was measured on 620 nm (Chameleon, Hidex Ltd., Turku, Finland) in three cultures of each experimental group in three independent experiments.

### 4.7. Zymography

Three-day-old cell cultures were harvested and homogenized in non-reducing homogenization buffer (50 mM TRIS base, 0.5% Triton X) at pH 7.4, containing protease inhibitors (Aprotinin (10 μg/mL), 5 mM Benzamidine, Leupeptin (10 μg/mL), Trypsine inhibitor (10 μg/mL), 1 mM PMSF, 5 mM EDTA, 1 mM EGTA, 8 mM Na-Fluoride, and 1 mM Na-orthovanadate). After centrifugation at 4 °C and 5000× *g*, supernatants were collected. Samples were stored at −70 °C. BCA protein assay (Pierce, Rockford, IL, USA) was used to set equal protein concentrations and about 20 µg of protein was separated by 8% SDS-PAGE gel. Gelatine solution (Sigma-Aldrich) for MMP9 activity detection, collagen type I solution (Sigma-Aldrich) for MMP9 and 13 activity detection, and casein solution (Sigma-Aldrich) for MMP1 activity detection were made and mixed in SDS gel. Samples were running at 90 V for 30 min. Gels were washed for 40 min in 500 mL 2.5% Triton X-100 solution (Amresco LLC, Solon, OH, USA) at room temperature, and then placed into 500 mL incubation buffer (1.47 g CaCl_2_·2H_2_O, 8.766 g NaCl, 6.057 g TRIS base, 0.5 g NaN_3_) at pH 8 for 20 h at 37 °C. Signals were visualized with shaking at 100 rpm for 2 h at room temperature in Coomassie Brillant Blue (Sigma-Aldrich) solution. Signals were detected with a gel documentary system (Fluorchem E, ProteinSimple, San Jose, CA, USA). An optical density of signals was measured using ImageJ 1.40 g freeware and results were normalized to that of untreated control or day zero cultures.

### 4.8. Aggrecanase Activity Assay

Aggrecan degradation is catalyzed by proteases of ADAMTS families, from which ADAMTS4 and ADAMTS5 cleave aggrecan. Aggrecanase Activity Assay measures the activity of ADAMTS in two steps; an aggrecanase part and an ELISA step. Briefly, an aggrecanase step, a recombinant fragment of human aggrecan interglobular domain (aggrecan-IGD), is digested with aggrecanase content of the samples. Proteolytic cleavage of the substrate releases an ARGSVIL-peptide, which can be quantified by the ELISA step. Assays were performed in the supernatant of the medium of chondrogenic cell cultures according to the protocol of manufacturers (MD Bioproducts, Zurich, Switzerland) after treatment with H_2_O_2_ and/or mechanical stress. After proteolytic digestion, the released aggrecan domains with ELISA modules were measured. Reactions were stopped by addition of sulphuric acid solution and absorbance was read at 450 nm in a microtiter plate spectrophotometer (Chameleon, Hidex Ltd., Turku, Finland).

### 4.9. Statistical Analysis

All data are representative of at least three independent experiments; data are mean values. Statistical significances between controls and PACAP treatments were determined by one-way analysis of variance (ANOVA), followed by Tukey’s HSD post-hoc test.

## Figures and Tables

**Figure 1 ijms-20-00168-f001:**
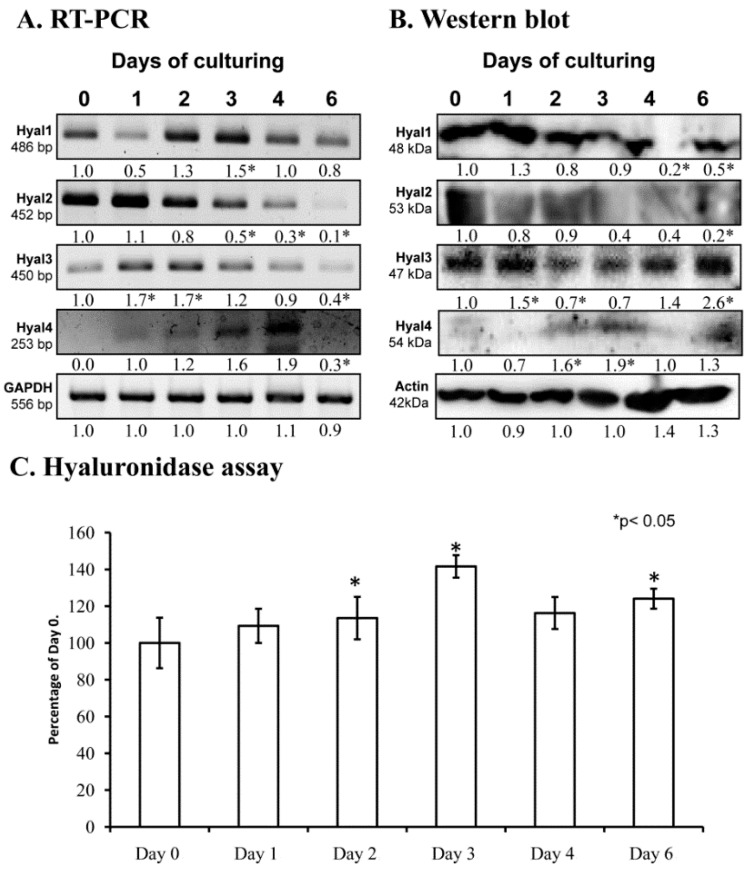
mRNA (**A**) and protein (**B**) expression of hyaluronidases in chondrifying micromass cultures. Optical densities of signals were measured and results were normalized to the optical densities of 0-day cultures. In panels (**A**,**B**), the numbers below the signals represent integrated densities of signals determined by ImageJ freeware. For reverse transcription followed by polymerase chain reactions (RT-PCR) and Western blot reactions, *glyceraldehyde 3-phosphate dehydrogenase** (GAPDH)* (**A**) or actin (**B**) were used as internal controls. Hyaluronidase activity in micromass cultures (**C**). Densities and means of three independent experiments (± standard error of the mean) are shown in the figures. Asterisks indicate significant differences compared with the 0-day cultures (* *p* < 0.05, one-way analysis of variance (ANOVA) followed by Tukey’s honestly significant difference (HSD) test).

**Figure 2 ijms-20-00168-f002:**
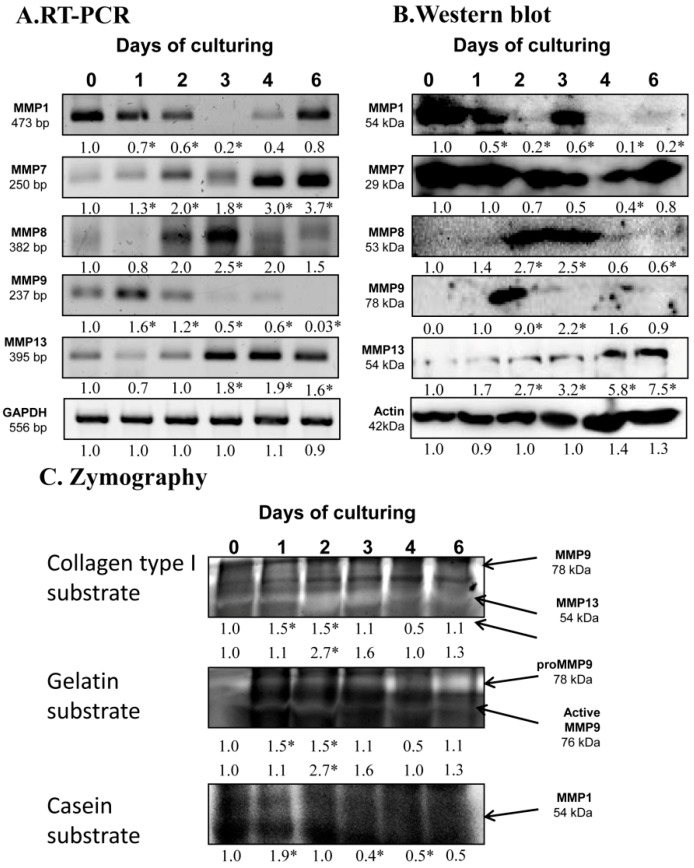
mRNA (**A**) and protein (**B**) expression of matrix metalloproteinases in chondrifying micromass cultures. Optical densities of signals were measured and results were normalized to the optical densities of 0-day cultures. In panels (**A**,**B**), the numbers below the signals represent integrated densities of signals determined by ImageJ freeware. For RT-PCR and Western blot reactions, *GAPDH* (**A**) and actin (**B**) were used as internal controls. Zymography (**C**) with collagen type I, gelatin, and casein substrates was also performed. Signals for MMP9 at 75 kDa, MMP13 at 54 kDa, proMMP9 at 85 kDa, and MMP1 at 54 kDa are labeled by arrows. Densities of three independent experiments are shown in the figures. Asterisks indicate significant differences compared to the 0-day cultures (* *p* < 0.05, one-way ANOVA followed by Tukey’s HSD test).

**Figure 3 ijms-20-00168-f003:**
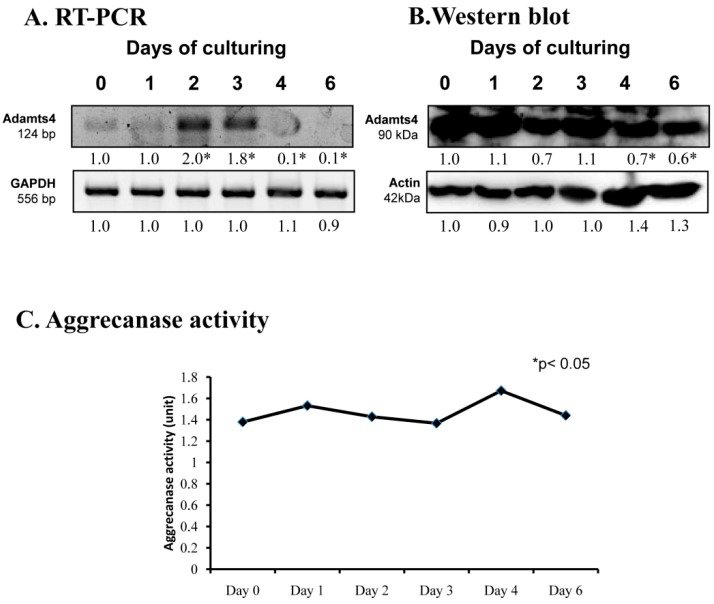
mRNA (**A**) and protein (**B**) expression of ADAMTS4 in chondrifying micromass cultures. Optical densities of signals were measured and results were normalized to the optical densities of 0-day cultures. In panels (**A**,**B**), the numbers below the signals represent integrated densities of signals determined by ImageJ freeware. For RT-PCR and Western blot reactions, *GAPDH* (**A**) and actin (**B**) were used as internal controls. Aggrecanase activity (**C**) was also determined in chondrifying micromass cell cultures. Densities and means of three independent experiments (±standard error of the mean) are shown in the figures. Asterisks indicate significant differences compared with the 0-day cultures (* *p* < 0.05, one-way ANOVA followed by Tukey’s HSD test).

**Figure 4 ijms-20-00168-f004:**
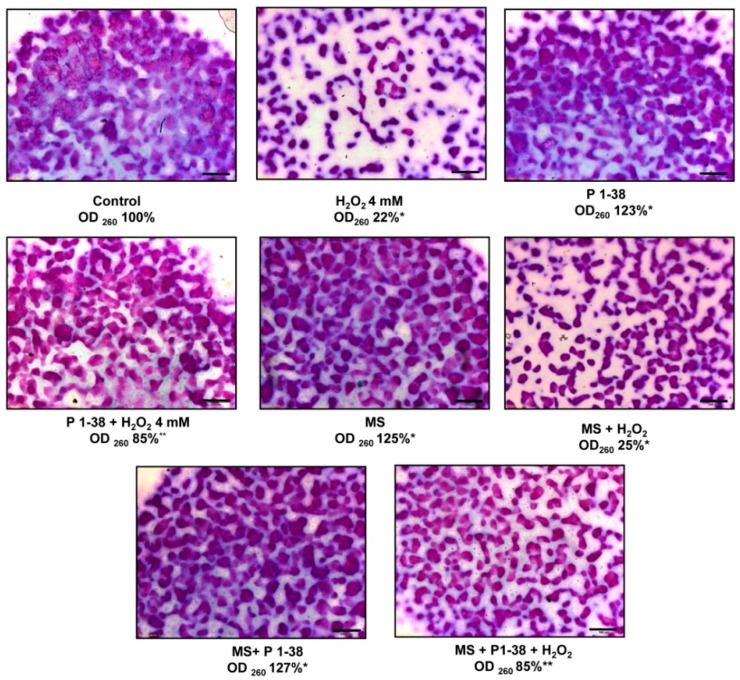
Effects of PACAP and/or oxidative stress and/or mechanical stress (MS) on metachromatic cartilage formation. Metachromatic cartilage areas in six-day-old cultures were visualized with DMMB dissolved in 3% acetic acid. Metachromatic (purple) structures represent cartilaginous nodules formed by many cells and a cartilage matrix rich in polyanionic GAG chains. Original magnification was 4×. Scale bar, 500 µm. Optical density (OD_625_) was determined in samples containing toluidine blue (TB) extracted with 8% HCl dissolved in absolute ethanol. Absorbance and means of three independent experiments (±standard error of the mean) are shown in the figures. Asterisks indicate significant differences compared with the six-day cultures (*p* < 0.05, one-way ANOVA followed by Tukey’s HSD test). * shows significant differences between Control cultures and treated samples; ** shows significant differences between single-H_2_O_2_ treatment and combined treatments.

**Figure 5 ijms-20-00168-f005:**
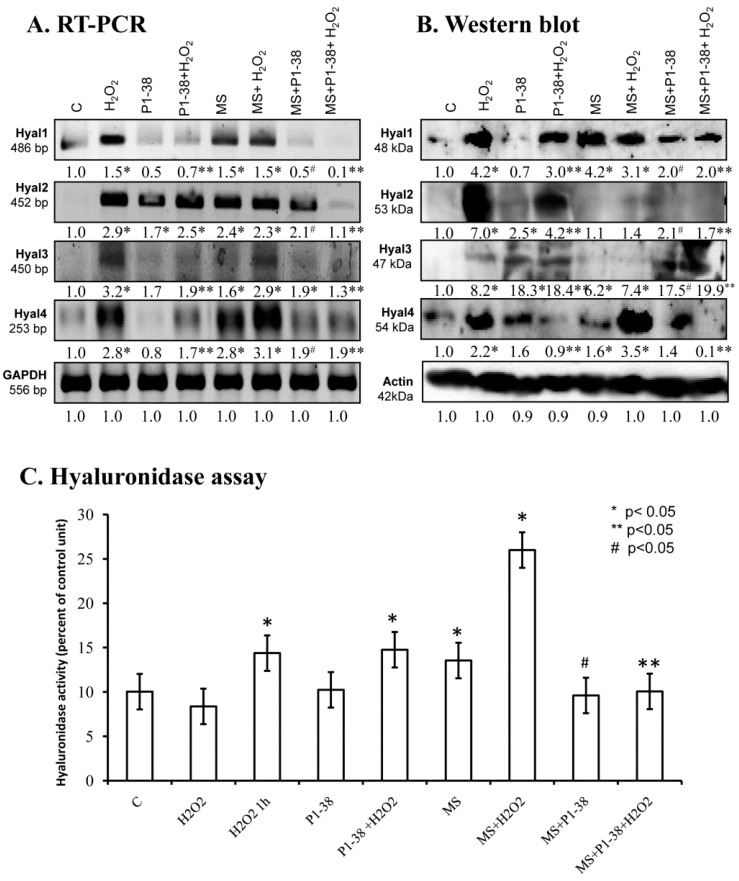
Effects of PACAP and/or oxidative stress and/or mechanical stress (MS) on mRNA (**A**) and protein (**B**) expression of hyaluronidases in chondrifying micromass cultures. Optical densities of signals were measured and results were normalized to the optical densities of three-day cultures (C). In panels (**A**,**B**), the numbers below the signals represent integrated densities of signals determined by ImageJ freeware. For RT-PCR and Western blot reactions, *GAPDH* (**A**) and actin (**B**) were used as internal controls. Hyaluronidase activity (**C**) was also determined in chondrifying micromass cell cultures during PACAP treatments, oxidative stress, mechanical stress, and combinations of the three. Densities and means of three independent experiments (±standard error of the mean) are shown in the figures. Asterisks indicate significant differences compared with the control cultures (*p* < 0.05, one-way ANOVA followed by Tukey’s HSD test). Where ANOVA reported significant differences between the groups (*p* < 0.05), a post hoc test (multiple comparison versus control group, Tukey’s HSD method) was used to isolate the groups that differed from the control group at *p* < 0.05. The respective control group was the untreated control when comparison was made between control, H_2_O_2_-treated, PACAP 1-38-treated, and mechanically stressed groups, whereas the H_2_O_2_-treated cultures were used as control in the post hoc test when comparison was made between H_2_O_2_-treated, H_2_O_2_+PACAP 1-38-treated, and H_2_O_2_+mechanically stressed samples. * shows significant differences between control cultures and treated samples; ** shows significant differences between single-H_2_O_2_ treatment and combined treatments; # shows significant differences between single-MS treatment and combined treatments.

**Figure 6 ijms-20-00168-f006:**
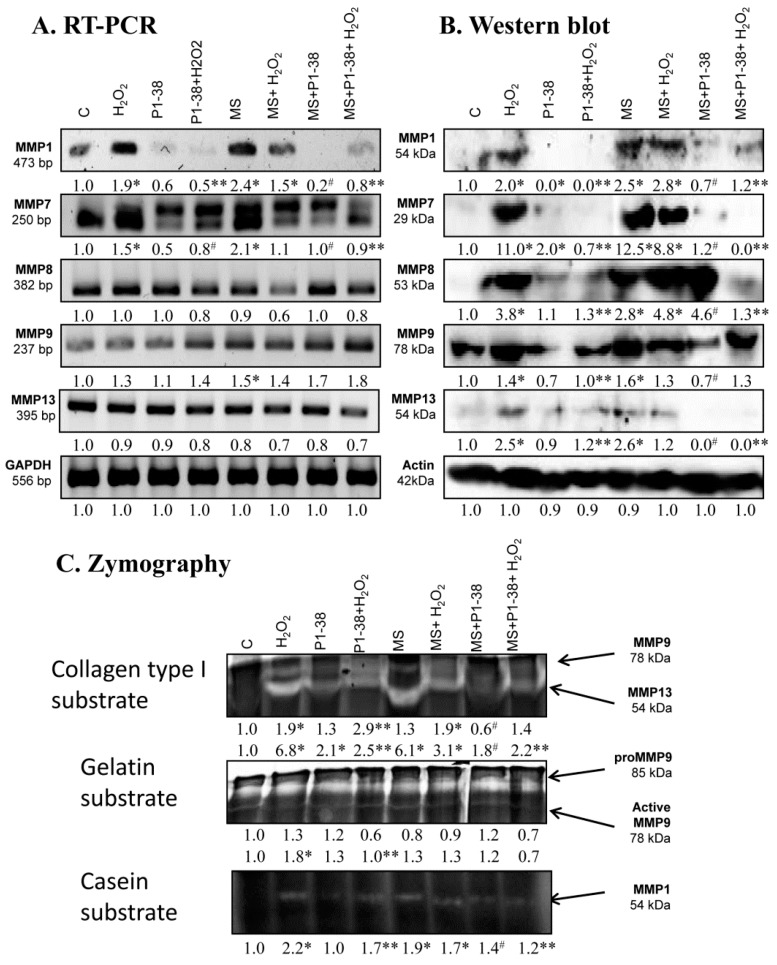
Effects of PACAP and/or oxidative stress and/or mechanical stress (MS) on mRNA (**A**) and protein (**B**) expression of matrix metalloproteinases in chondrifying micromass cultures. Optical densities of signals were measured and results were normalized to the optical densities of three-day cultures (**C**). In panels (**A**,**B**), the numbers below the signals represent integrated densities of signals determined by ImageJ freeware. For RT-PCR and Western blot reactions, *GAPDH* (**A**) and actin (**B**) were used as internal controls. Zymography (**C**) with collagen type I, gelatin, and casein substrates was also performed during PACAP treatments, oxidative stress, mechanical stress, and combinations of the three. Signals for MMP9 at 75 kDa, MMP13 at 54 kDa, proMMP9 at 85 kDa, and MMP1 at 54 kDa are labeled. Densities and means of three independent experiments (±standard error of the mean) are shown in the figures. Asterisks indicate significant differences compared with the three-day cultures (*p* < 0.05, one-way ANOVA followed by Tukey’s HSD test). Where ANOVA reported significant differences among the groups (*p* < 0.05), a post hoc test (multiple comparison versus control group, Tukey’s HSD method) was used to isolate the groups that differed from the control group at *p* < 0.05. The respective control group was the untreated control when comparison was made between control, H_2_O_2_-treated, PACAP 1-38-treated, and mechanically stressed groups, whereas the H_2_O_2_-treated cultures were used as control in the post hoc test when comparison was made between H_2_O_2_-treated, H_2_O_2_+PACAP 1-38-treated, and H_2_O_2_+mechanically stressed samples. * shows significant differences between control cultures and treated samples; ** shows significant differences between single-H_2_O_2_ treatment and combined treatments; # shows significant differences between single-MS treatment and combined treatments.

**Figure 7 ijms-20-00168-f007:**
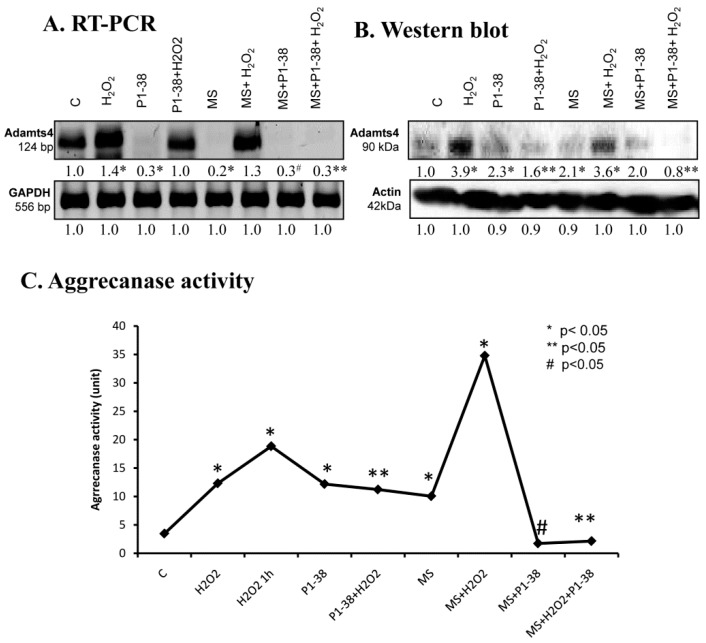
Effects of PACAP and/or oxidative stress and/or mechanical stress (MS) on mRNA (**A**) and protein (**B**) expression of ADAMTS4 in chondrifying micromass cultures. Optical densities of signals were measured and results were normalized to the optical densities of three-day cultures (**C**). In panels (**A**,**B**), the numbers below the signals represent integrated densities of signals determined by ImageJ freeware. For RT-PCR and Western blot reactions, *GAPDH* (**A**) and actin (**B**) were used as internal controls. Aggrecanase activity (**C**) was also measured after treatments. Densities and means of three independent experiments (±standard error of the mean) are shown in the figures. Asterisks indicate significant differences compared to the three-day cultures (*p* < 0.05, one-way ANOVA followed by Tukey’s HSD test). Where ANOVA reported significant differences between the groups (*p* < 0.05), a post hoc test (multiple comparison versus control group, Tukey’s HSD method) was used to isolate the groups that differed from the control group at *p* < 0.05. The respective control group was the untreated control when comparison was made between control, H_2_O_2_-treated, PACAP 1-38-treated, and mechanically stressed groups, whereas the H_2_O_2_-treated cultures were used as control in the post hoc test when comparison was made between H_2_O_2_-treated, H_2_O_2_+PACAP 1-38-treated, and H_2_O_2_+mechanically stressed samples. * shows significant differences between control cultures and treated samples; ** shows significant differences between single-H_2_O_2_ treatment and combined treatments; # shows significant differences between single-MS treatment and combined treatments.

**Table 1 ijms-20-00168-t001:** Nucleotide sequences, amplification sites, GenBank accession numbers, amplimer sizes, and polymerase chain reaction(PCR) reaction conditions for each primer pair are shown.

Gene	Primer	Nucleotide Sequence (5’→3’)	GenBank ID	Annealing Temperature	Amplimer Size (bp)
*Admats4*	sense	GTG GCA AGT ATT GTG AGG G(2079–2097)	NM_172845.3	54 °C	124
	antisense	AGG TCG GTT CGG TGG TT(2186–2202)			
*Hyal1*	sense	GGG GTC TTT GAT GTC GTG G(371–389)	XM_015292793.1	57 °C	486
	antisense	CGG GTC GCT GAA GTT GTT(839–856)			
*Hyal2*	sense	ACA ACC ACG ACT ACA GCA AGA A(784–805)	XM_414258.5	56 °C	452
	antisense	CGC TGC CAT CGT CAC ATT(1218–1235)			
*Hyal3*	sense	TAC GGC ATC GTG GAG AAC CG(265–284)	XM_003641994.3	59 °C	450
	antisense	CCA GTC GTC GTT GAA GCA GTC G(693–714)			
*Hyal4*	sense	CCA CCG TGC CTT GCT ATT(325–342)	XM_017011911.1	51 °C	254
	antisense	GTT TGC TGC TGG TCC TTT(560–577)			
*MMP1*	sense	TTT GTG ACC CTA ACT TGA(1021–1038)	XM_417176.4	47 °C	473
	antisense	GAC ATA GCC ATC TTT CTG(1476–1493)			
*MMP7*	sense	AAA AGA GTT ACC TCG GGA CA(483–502)	NM_007742.3	52 °C	250
	antisense	CAC GGA CAT TTG AGT GGG(717–735)			
*MMP8*	sense	TGT CAA GGG CTG AAG TGA(478–495)	NM_008611.4	51 °C	382
	antisense	TGA GGT AGT GAA TAG GTG C(841–859)			
*MMP9*	sense	TTC TGG ACT CTG GGA ACC G(625–643)	NM_204667.1	57 °C	237
	antisense	GGG AGA CCC ATC GCT GTT(844–861)			
*MMP13*	sense	CAT GCA GAA ACC ACG ATG(296–313)	NM_001293090.1	51 °C	395
	antisense	GAG CAG CAA CAA GAA ACA AG(671–690)			
*GAPDH*	sense	GAG AAC GGG AAA CTT GTC AT(238–258)	NM_204305	54 °C	556
	antisense	GGC AGG TCA GGT CAA CAA(775–793)			

**Table 2 ijms-20-00168-t002:** Tables of antibodies used in the experiments.

Antibody	Host Animal	Dilution	Distributor
Anti-Hyal1	rabbit, polyclonal	1:500	Sigma-Aldrich, St. Louis, MO, USA
Anti-Hyal2	rabbit, polyclonal	1:500	Sigma-Aldrich, St. Louis, MO, USA
Anti-Hyal3	rabbit, polyclonal	1:500	Sigma-Aldrich, St. Louis, MO, USA
Anti-Hyal4	rabbit, polyclonal	1:500	Sigma-Aldrich, St. Louis, MO, USA
Anti-MMP1	rabbit, polyclonal	1:500	Sigma-Aldrich, St. Louis, MO, USA
Anti-MMP7	rabbit, polyclonal	1:500	Sigma-Aldrich, St. Louis, MO, USA
Anti-MMP8	rabbit, polyclonal	1:500	Sigma-Aldrich, St. Louis, MO, USA
Anti-MMP9	rabbit, polyclonal	1:500	Sigma-Aldrich, St. Louis, MO, USA
Anti-MMP13	rabbit, polyclonal	1:500	Sigma-Aldrich, St. Louis, MO, USA
Anti-Adamts4	rabbit, polyclonal	1:500	Sigma-Aldrich, St. Louis, MO, USA
Anti-Actin	mouse, monoclonal	1:10,000	Sigma-Aldrich, St. Louis, MO, USA

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
