# Peer review of "Pituitary Adenylate Cyclase Activating Polypeptide (PACAP) Reduces Oxidative and Mechanical Stress-Evoked Matrix Degradation in Chondrifying Cell Cultures"

_ijms, 2019, doi:10.3390/ijms20010168_

Round 1

Reviewer 1 Report

Summary:

The manuscript describes the action of metalloproteinases, including hyaluronidases, MMPs and an aggreacanase in chondrogenesis in primary chicken micromass. Moreover, the authors study the consequences of the application of two degradative mediators, oxidative and mechanical stress, in the expression and activity of these metalloproteinases, as well as the effect of the neuropeptide PACAP. PACAP seems to have a positive effect by the inhibition of matrix degrading enzymes. In general, the manuscript is well written and would be useful to the discovery of new therapeutic methods based on neuropeptides for the treatment of cartilage degrading pathologies such as osteoarthritis. However, some experimentation and minor points need to be improved before reaching any conclusion.

Broad comments:

1.    Western blots along the manuscript are useful to see the effects at protein level and to be able to compare with the mRNA transcripts. However, Western blots should be improved since it is very difficult to observe the protein bands and the differences between them (e.g. figures 1B, 2B, 5B). Moreover, some bands are very close to be able to quantify them properly (e.g. figure 2: MMP7, MMP8; figure 3: ADAMTS4).

2.    In nearly all cases, mRNA expression does not correlate with the protein. This fact should be more broadly discussed.

Specific comments:

1.    Page 4, line 119. Authors state that “mRNA expression of MMP8 became stronger from days of differentiation”. However, at days 4 and 6 its expression decreases.

2.    Page 4, line 128. Authors say “As the expression pattern of these enzymes is not always in correlation with their activity…” I suppose that they wanted to mean protein instead of activity. It should be corrected.

3.    Page 6, lines 191-192, 194-195, 196-197. Authors affirm that “The combined application of the two stress elevated the protein expression of all hyaluronidases”, It is difficult to see in Hyal2 and 3; “PACAP also prevented the harmful effect of mechanical stress and lowered the protein expression of Hyal1 and Hyal4”, It is difficult to see in Hyal4; “…in the presence of PACAP and stress factors…protein signals of Hyal2 and Hyal3 became stronger”, It is difficult to see in Hyal2. As suggested in the broad comment 1., Western blots should be improved.

4.    Page 8, line 234. Please correct MPPs.

5.    Page 8, lines 228-229 and 235-236. Authors state that mRNA expression of MMP9 increased in the presence of PACAP and mechanical load, while the protein expression diminished. Why?

6.    Authors measure the mRNA and protein expression of ADAMTS4 as well as de aggrecanases activity. However this activity is also due to the action of other aggrecanases such as ADAMTS5. This ADAMTS should also be studied or authors should discuss it.

7.    Page 10, line 281. “…resulted in a double fold elevation in the activity of ADAMTS protein”Activity should be replaced by expression.

8.    Page 10, lines 284-285. As described in 6., aggrecanases activity should not be exclusively attributed to ADAMTS4.

9.    Page 10, lines 286-287. According to the figure 7C, PACAP reduced the aggrecanase activity to control level in the presence of mechanical loading or after the combined application of the two stresses, not in the presence of H2O2 as described here.

10.  Page 11, lines 324-331. Authors state that “…full activity of hyaluronidases was increased on days of chondrogenic differentiation…” However, authors also say that “Hyal4 does not have hyaluronidase activity”, and “…Hyal2 and Hyal3 protein expression reduced on days of chondrogenic differentiation while Hyal4 elevated and Hyal1 did not alter on these days”. Which are the hyaluronidases responsible of that hyaluronidase activity increase?

11.  Page 13, line 414. Authors suggest an MMP8 inhibition by PACAP. However, it is neither observed at mRNA, nor at protein level (Figure 6A, B). It is only seen in the protein expression after treatment with H2O2 compared to H2O2 alone.

12.  Page 13, lines 421-422. Authors state that “oxidative stress also increased the expression and the activation of MMP9, which was normalized by PACAP addition.” Which is true at protein level but not with the mRNA. This fact should be discussed.

13.  Figures 5, 6 and 7.

13.1.     As indicated in the figure legends “results were normalized to the optical densities of 0-day cultures”. Shouldn’t they be normalized to the control cultures?

13.2.     Figures 6 and 7. “Asterisks indicate significant differences compared to the 0-day cultures”. Shouldn’t they be compared to the control cultures as in figure 5?

13.3.     Figures 5 and 7. *p < 0,05 and **p < 0.05 is indicated in the figure. Shouldn’t they be *p < 0.05 and **p < 0.001. It would be better to indicate it in the figure legends.

13.4.     Symbols should be better described in the figure legends: *, **, #.

Author Response

Summary:

The manuscript describes the action of metalloproteinases, including hyaluronidases, MMPs and an aggreacanase in chondrogenesis in primary chicken micromass. Moreover, the authors study the consequences of the application of two degradative mediators, oxidative and mechanical stress, in the expression and activity of these metalloproteinases, as well as the effect of the neuropeptide PACAP. PACAP seems to have a positive effect by the inhibition of matrix degrading enzymes. In general, the manuscript is well written and would be useful to the discovery of new therapeutic methods based on neuropeptides for the treatment of cartilage degrading pathologies such as osteoarthritis. However, some experimentation and minor points need to be improved before reaching any conclusion.

Broad comments:

1.    Western blots along the manuscript are useful to see the effects at protein level and to be able to compare with the mRNA transcripts. However, Western blots should be improved since it is very difficult to observe the protein bands and the differences between them (e.g. figures 1B, 2B, 5B). Moreover, some bands are very close to be able to quantify them properly (e.g. figure 2: MMP7, MMP8; figure 3: ADAMTS4).

During the analysis we tried to increase the quality of the bands by using Supersignal West Femto kit (Pierce TM, MA, USA) but the background of the signals became extremely strong. We have tried to improve the Western blots with better blocking or higher protein concentration without better results, therefore, we repeated the experiments 3 times and we made statistical analysis of the data obtained from these 3 independent experiments. In every experiment we made density analysis and normalized the results on its own internal control. Then the 3 independent results were compared and statistical analysis has been done. In the Figures we presented data of the experiment out of the three which best represented the average results. We agree with the reviewer that these blots could be better, but despite of our efforts to improve the quality of those, we did not succeed in that. We suppose that the source of the mixed nature of the cell population as it comes from "high density" cell cultures from chicken limb buds can be the reason why we saw the decreased blot intensity.    

2.    In nearly all cases, mRNA expression does not correlate with the protein. This fact should be more broadly discussed.

Chondrogenic high density cultures is a rapidly transforming system, where the mRNA expression dynamically changes and timing of investigation is not always makes it possible to see correlation between mRNA and protein expression (Zakany et al., 2005). Moreover, we were not surprised to get these results as previously in PACAP KO mice femur (Jozsa et al., 2018) or in testis (Fulop et al., 2018) or in amyloidosis (Reglodi et al., 2018) Western blot and mRNA levels were not in correlation. In UMR-106 cell line, PACAP did not alter mRNA expression but protein expression was changed (Juhasz et al., 2014). In high density cell cultures, addition of PACAP did not significantly modulate the mRNA expression of certain matrix producing enzymes or matrix proteins (Juhasz et al., 2014, 2015). These findings indicate, that PACAP-signaling exerts more pronouncing effect on posttranscriptional events and protein metabolism of cells and it has less influence on gene expression of the targeted downstream molecules.

Zákány R, Szíjgyártó Z, Matta C, Juhász T, Csortos C, Szucs K, Czifra G, Bíró T, Módis L, Gergely P. Hydrogen peroxide inhibits formation of cartilage in chicken micromass cultures and decreases the activity of calcineurin: implication of ERK1/2 and Sox9 pathways. Exp Cell Res. 2005 Apr 15;305(1):190-9.

Józsa G, Szegeczki V, Pálfi A, Kiss T, Helyes Z, Fülöp B, Cserháti C, Daróczi L, Tamás A, Zákány R, Reglődi D, Juhász T. Signalling Alterations in Bones of Pituitary Adenylate Cyclase Activating Polypeptide (PACAP) Gene Deficient Mice. Int J Mol Sci. 2018 Aug 27;19(9).

Fulop BD, Sandor B, Szentleleky E, Karanyicz E, Reglodi D, Gaszner B, Zakany R, Hashimoto H, Juhasz T, Tamas A. Altered Notch Signaling in Developing Molar Teeth of Pituitary Adenylate Cyclase-Activating Polypeptide (PACAP)-Deficient Mice. J Mol Neurosci. 2018 Aug 10. doi: 10.1007/s12031-018-1146-7.

Reglodi D, Jungling A, Longuespée R, Kriegsmann J, Casadonte R, Kriegsmann M, Juhasz T, Bardosi S, Tamas A, Fulop BD, Kovacs K, Nagy Z, Sparks J, Miseta A, Mazzucchelli G, Hashimoto H, Bardosi A. Accelerated pre-senile systemic amyloidosis in PACAP knockout mice - a protective role of PACAP in age-related degenerative processes. J Pathol. 2018 Aug;245(4):478-490.

Juhász T, Matta C, Katona É, Somogyi C, Takács R, Hajdú T, Helgadottir SL, Fodor J, Csernoch L, Tóth G, Bakó É, Reglődi D, Tamás A, Zákány R. Pituitary adenylate cyclase-activating polypeptide (PACAP) signalling enhances osteogenesis in UMR-106 cell line.

J Mol Neurosci. 2014 Nov;54(3):555-73.

Juhász T, Matta C, Katona É, Somogyi C, Takács R, Gergely P, Csernoch L, Panyi G, Tóth G, Reglődi D, Tamás A, Zákány R. Pituitary adenylate cyclase activating polypeptide (PACAP) signalling exerts chondrogenesis promoting and protecting effects: implication of calcineurin as a downstream target. PLoS One. 2014 Mar 18;9(3):e91541.

Juhász T, Szentléleky E, Somogyi CS, Takács R, Dobrosi N, Engler M, Tamás A, Reglődi D, Zákány R. Pituitary Adenylate Cyclase Activating Polypeptide (PACAP) Pathway Is Induced by Mechanical Load and Reduces the Activity of Hedgehog Signaling in Chondrogenic Micromass Cell Cultures. Int J Mol Sci. 2015 Jul 29;16(8):17344-67.

Specific comments:

1.       Page 4, line 119. Authors state that “mRNA expression of MMP8 became stronger from days of differentiation”. However, at days 4 and 6 its expression decreases.

Sentence was modified as follows: "mRNA expression of MMP8 became strong on the days of chondrogenic transformation, then decreased by the end of culturing."

2.       Page 4, line 128. Authors say “As the expression pattern of these enzymes is not always in correlation with their activity.” I suppose that they wanted to mean protein instead of activity. It should be corrected.

Sentence was corrected: "As the protein expression pattern of these enzymes is not always in correlation with their catalytic activity."

3.       Page 6, lines 191-192, 194-195, 196-197. Authors affirm that “The combined application of the two stress elevated the protein expression of all hyaluronidases”, It is difficult to see in Hyal2 and 3; “PACAP also prevented the harmful effect of mechanical stress and lowered the protein expression of Hyal1 and Hyal4”, It is difficult to see in Hyal4; “…in the presence of PACAP and stress factors…protein signals of Hyal2 and Hyal3 became stronger”, It is difficult to see in Hyal2. As suggested in the broad comment 1., Western blots should be improved.

We agree with the reviewer that these sentences are confusing, therefore, we rephrased them:

"The combined application of the two stress factors elevated the expression of Hyal1, Hyal3 and Hyal4."

"PACAP did not completely prevent the harmful effect of mechanical stress and did not significantly lower the protein expression of Hyal1 and Hyal4 (Fig. 5B)."

"Moreover, the protein expression of Hyal1 and Hyal4 decreased in the presence of PACAP and stress factors, but protein signals of Hyal3 became stronger (Fig. 5B). No significant changes were visible in Hyal2 (Fig. 5B)."

4.       Page 8, line 234. Please correct MPPs.

The typing mistake has been corrected.

5.    Page 8, lines 228-229 and 235-236. Authors state that mRNA expression of MMP9 increased in the presence of PACAP and mechanical load, while the protein expression diminished. Why?

We apologize the Reviewer for the confusing phrasing of the results describing the expression of MMP9. In case of this enzyme both the mRNA and protein expressions increased after the application of stress factors. We corrected the text accordingly.

In case of MMP8 and MMP13 the protein expression changed differently than the mRNA expression, protein levels increased after the various stress, while MMP9 protein responded similarly to mRNA expression (Fig. 6B).”  

6.    Authors measure the mRNA and protein expression of ADAMTS4 as well as de aggrecanases activity. However this activity is also due to the action of other aggrecanases such as ADAMTS5. This ADAMTS should also be studied or authors should discuss it.

We completely agree with the Reviewer that both ADAMTS have important function in arthritis formation. Therefore, ADAMTS5 function can also be an important target in treatment of osteoarthritis. Although both ADAMTS4 and ADMATS5 have the same predominant function to cleave aggrecane. But we considered that ADAMTS4 has a direct connection with Sox9 (Zhang et al., 2015) which activity is regulated partly by PACAP (Juhasz et al., 2014). This is the reason why we investigated only the ADAMTS4 function.

Zhang Q1, Ji QWang XKang LFu YYin YLi ZLiu YXu XWang Y. SOX9 is a regulator of ADAMTSs-induced cartilage degeneration at the early stage of human osteoarthritis. Osteoarthritis Cartilage. 2015 Dec;23(12):2259-2268.

Juhász T, Matta C, Katona É, Somogyi C, Takács R, Gergely P, Csernoch L, Panyi G, Tóth G, Reglődi D, Tamás A, Zákány R. Pituitary adenylate cyclase activating polypeptide (PACAP) signalling exerts chondrogenesis promoting and protecting effects: implication of calcineurin as a downstream target. PLoS One. 2014 Mar 18;9(3):e91541.

7.    Page 10, line 281. “…resulted in a double fold elevation in the activity of ADAMTS protein”Activity should be replaced by expression.

Sentence was modified as follows: "Surprisingly, PACAP as well as mechanical load resulted in a double fold elevation in the expression of ADAMTS protein."

8.    Page 10, lines 284-285. As described in 6., aggrecanases activity should not be exclusively attributed to ADAMTS4.

We corrected the sentence: “PACAP and mechanical load similarly to the protein expression resulted in an elevation of ADAMTS with aggrecanase activity (Fig. 7C).”

9.    Page 10, lines 286-287. According to the figure 7C, PACAP reduced the aggrecanase activity to control level in the presence of mechanical loading or after the combined application of the two stresses, not in the presence of H2O2 as described here.

We corrected the sentence: "PACAP reduced the aggrecanase activity to control level in the presence of mechanical loading and after the combined application of the two stress. On the other hand, ADAMTS activity responded with moderate decrease only to the PACAP treatment when oxidative stress was applied without mechanical stress. (Fig. 7C)."

10.  Page 11, lines 324-331. Authors state that “…full activity of hyaluronidases was increased on days of chondrogenic differentiation…” However, authors also say that “Hyal4 does not have hyaluronidase activity”, and “…Hyal2 and Hyal3 protein expression reduced on days of chondrogenic differentiation while Hyal4 elevated and Hyal1 did not alter on these days”. Which are the hyaluronidases responsible of that hyaluronidase activity increase?

We apologize for the confusing sentence, therefore we rephrased it.  „Our data show that activity of hyaluronidases was increased on days of chondrogenic differentiation suggesting the importance of a dynamic change of extracellular hyaluronan content during commitment of chondroprogenitor cells.”

According to our data Hyal2 and Hyal3 are the main hyaluronidases which play key role in micromass chondrogenic cell differentiation.

11.  Page 13, line 414. Authors suggest an MMP8 inhibition by PACAP. However, it is neither observed at mRNA, nor at protein level (Figure 6A, B). It is only seen in the protein expression after treatment with H2O2 compared to H2O2 alone.

According to the suggestion of the reviewer we modified the sentence: "Our results suggest that inhibition of MMP8 protein expression by PACAP in oxidative stress normalizes and keeps in balance the cartilage matrix production."

12.  Page 13, lines 421-422. Authors state that “oxidative stress also increased the expression and the activation of MMP9, which was normalized by PACAP addition.” Which is true at protein level but not with the mRNA. This fact should be discussed.

Thank you for the suggestion of the Reviewer. We added discussing sentences to the text in lines 443-446.

„As the mRNA expression of MMP9 remained unaffected in oxidative stress, either with or without PACAP addition while protein expression changed in both cases, it suggests that PACAP influences MMP9 activity without modulation its gene expression.”

13.  Figures 5, 6 and 7.

13.1.     As indicated in the figure legends “results were normalized to the optical densities of 0-day cultures”. Shouldn’t they be normalized to the control cultures?

We apologize the reviewer for this mistake in the Legends, data obtained from the samples of untreated control cultures of the same age (i.e. day 3) were applied as reference. We corrected the statement of the Legends.

13.2.     Figures 6 and 7. “Asterisks indicate significant differences compared to the 0-day cultures”. Shouldn’t they be compared to the control cultures as in figure 5?

In stress application samples from day 3 of culturing were considered as control cultures. We corrected the Figure legends in Figure 5, 6 and 7.

13.3.     Figures 5 and 7. *p < 0,05 and **p < 0.05 is indicated in the figure. Shouldn’t they be *p < 0.05 and **p < 0.001. It would be better to indicate it in the figure legends.

*p < 0.05 shows the significant differences between Control cultures and treated groups. **p < 0.05 shows the significant differences between H2O2 single treatment and combined treatments.

13.4.     Symbols should be better described in the figure legends: *, **, #.

* shows significant differences between Control cultures and treated samples.

** shows significant differences between single-H2O2 treatment and combined treatments.

# shows significant differences between single-MS treatment and combined treatments.

Reviewer 2 Report

In the present publication, the authors investigated the impact of the neuropeptide PACAP on the degradation of the main components of the extracellular matrix in chicken limb micromass cultures treated with H2O2 and/or mechanical stress. In the first part the expression and activity of hyaluronidases, MMPs and Adamts4 in micromass cultures were analyzed. In the second part, the authors analyzed the expression and activity of the enzymes, which are responsible for the degradation of the ECM, in presence of cellular stress and PACAP. The addition of PACAP reduced the degradation of the ECM by cellular stress.

Altogether the authors were able to show the potential of PACAP for the treatment of osteoarthritis and rheumatoid arthritis. However for the publication in International Journal of Molecular Science, I suggest some additional experiments.

The quality of the images should be improved. In the figures the neuropeptide should be named consistent as PI-38 or as PACAP. There are some spelling errors (page 8 line 234 MPPs instead of MMPs; page 11 line 338: chondorcytes). The numeration in the results part is wrong.

The results are contrary on mRNA- and protein level. The authors explain these discrepancies by putative regulationof the translation. In my opinion this is not convincing. First, the qualities of the blots are not always suitable to quantify the signal intensity by ImageJ. So far I know, Sigma Aldrich offer no antibodies with specifity against chicken. The specifity of the antibodies for the different isoforms may be even not sure. Therefore the authors should evaluate the antibody specifity by negative controls (e.g. siRNA experiments). Furthermore analysis of the mRNA expression levels by qPCR would be helpful.There is no correlation between the expression levels, especially the protein levels, and the activity. In the hyaluronidase assay, I would expect higher activity on day 6 of the untreated micromass cultures because of the intensive band in the HYAL3 western blot. Also in H2O2 treated samples the activity of the Hyal should be higher. Here, a second method should be done (e.g. zymography; ELISA).

Analysis of the expression levels of the chondrogenic markers for confirmation of the differentiation stages (e.g. type II collagen, Indian hedgehog and type X collagen)and of the enzymes forming the ECM would verify the conclusion of the study.

Author Response

Comments and Suggestions for Authors

In the present publication, the authors investigated the impact of the neuropeptide PACAP on the degradation of the main components of the extracellular matrix in chicken limb micromass cultures treated with H2O2 and/or mechanical stress. In the first part the expression and activity of hyaluronidases, MMPs and Adamts4 in micromass cultures were analyzed. In the second part, the authors analyzed the expression and activity of the enzymes, which are responsible for the degradation of the ECM, in presence of cellular stress and PACAP. The addition of PACAP reduced the degradation of the ECM by cellular stress.

Altogether the authors were able to show the potential of PACAP for the treatment of osteoarthritis and rheumatoid arthritis. However for the publication in International Journal of Molecular Science, I suggest some additional experiments.

       The quality of the images should be improved. In the figures the neuropeptide should be named consistent as PI-38 or as PACAP. There are some spelling errors (page 8 line 234 MPPs instead of MMPs; page 11 line 338: chondorcytes). The numeration in the results part is wrong.

We modified the figures and we used P1-38. Spelling mistakes and the corresponding numeration are corrected in the entire text .

2.               The results are contrary on mRNA- and protein level. The authors explain these discrepancies by putative regulation of the translation. In my opinion this is not convincing. First, the qualities of the blots are not always suitable to quantify the signal intensity by ImageJ. So far I know, Sigma Aldrich offer no antibodies with specifity against chicken. The specifity of the antibodies for the different isoforms may be even not sure. Therefore the authors should evaluate the antibody specifity by negative controls (e.g. siRNA experiments). Furthermore analysis of the mRNA expression levels by qPCR would be helpful. There is no correlation between the expression levels, especially the protein levels, and the activity. In the hyaluronidase assay, I would expect higher activity on day 6 of the untreated micromass cultures because of the intensive band in the HYAL3 western blot. Also in H2O2 treated samples the activity of the Hyal should be higher. Here, a second method should be done (e.g. zymography; ELISA).

We agree with the Reviewer that real-time PCR would be a more adequate method for quantitative analysis. We have made a lot of efforts to isolate pure RNA from HD cultures without great success. We used kits and classical isolation methods and the purity of RNA was not good enough to run reliable Q-PCR reactions. We observed the protein expression of the investigated molecules with Western blot method, in most of the cases, were not in good correlation with the mRNA expression we presented. As we have repeated the experiments at least three times and statistical analysis was performed, therefore, we believe that the results obtained with this semi-quantitative method serve as a good starting point to understand PACAP effect on gene expression in micromass cultures. In Western blot analysis we tried to increase the quality of the signal by using Supersignal West Femto kit (PierceTM, MA, USA) but the background of the signals became extremely strong. Therefore, we repeated the Western blot reactions at least 3 times, and we made statistical analysis. These results led us to our conclusions.

We completely agree with the reviewer that Sigma or other distributors do not provide antibodies with specificity to chicken species. Therefore, we used polyclonal antibodies which recognize the evolutionary conserved part of the target proteins and we investigated only those bands which appear in the molecular weight shown by the data sheet. In our previous experiments we have tried to use siRNA constructs to prove the specificity of the antibodies (Juhasz et al., 2014). Silencing of PKA and PP2B was not possible in micromass cell cultures. The transfection of siRNA to HD cultures gave us very low silencing results with Lipofectamine method or other Liposome based transfections. On the other hand, electroporation resulted a high gene silencing ratio but it caused severe cell death and we lost the high density characteristic of micromass cultures required for chondrogenesis. In these cases although the gene silencing gave a good result but cultures do not differentiate to cartilage anymore because of the low density of cells. In case of lost high density characteristic cartilage specific mRNA and protein expressions almost disappeared.

Hyaluronidase assay results were partly surprising for us as well, but we have similar results with the activity of PP2B or ALP previously (Juhasz et al., 2014 and Jozsa et al., 2018). The increased expression of these proteins do not always in correlation with their activity or sometimes parallel to an unchanged protein level of an enzyme such as PP2A an elevated enzyme activity was assayed (Reglodi et al., 2018). These results support that PACAP has several signaling crosstalks which can either influence the expression of molecules or increase their activity without any change of protein levels. Hyal activity on day 6 of culturing should not be increased in high density cultures as by that day mature cartilage is found in the cultures. Further development of these cultures leads to a hypertrophic transformation of cells and beginning of calcification in the formed hyaline cartilage. We suppose that this elevation of Hyal3 expression can be a first sign of this process as the enzyme may remove hyaluronic acid from the cartilage matrix which is essential for calcium salt deposition (Tanimoto et al., 2004). Application of H2O2 can induce several other processes such as apoptosis and necrosis, we used such concentration of this agent which induces only slight apoptotic or necrotic processes (Juhasz et al., 2014) (see in Supplementary Figure), which can partly explain the elevated Hyal3 expression (Tanimoto et al., 2004). Moreover, there are data that elevated protein expression of an enzyme is not necessarily parallel with increase its overall assayable catalytic activity. It is published that non-specific complexes between HA fragments and Hyals themselves inhibited the Hyals catalytic activity towards HA (Deschrevel et al., 2008).  

Juhász T, Matta C, Somogyi C, Katona É, Takács R, Soha RF, Szabó IA, Cserháti C, Sződy R, Karácsonyi Z, Bakó E, Gergely P, Zákány R. Mechanical loading stimulates chondrogenesis via the PKA/CREB-Sox9 and PP2A pathways in chicken micromass cultures. Cell Signal. 2014 Mar;26(3):468-82.

Józsa G, Szegeczki V, Pálfi A, Kiss T, Helyes Z, Fülöp B, Cserháti C, Daróczi L, Tamás A, Zákány R, Reglődi D, Juhász T. Signalling Alterations in Bones of Pituitary Adenylate Cyclase Activating Polypeptide (PACAP) Gene Deficient Mice. Int J Mol Sci. 2018 Aug 27;19(9).

Juhász T, Matta C, Katona É, Somogyi C, Takács R, Gergely P, Csernoch L, Panyi G, Tóth G, Reglődi D, Tamás A, Zákány R. Pituitary adenylate cyclase activating polypeptide (PACAP) signalling exerts chondrogenesis promoting and protecting effects: implication of calcineurin as a downstream target. PLoS One. 2014 Mar 18;9(3):e91541.

Reglodi D, Cseh S, Somoskoi B, Fulop BD, Szentleleky E, Szegeczki V, Kovacs A, Varga A, Kiss P, Hashimoto H, Tamas A, Bardosi A, Manavalan S, Bako E, Zakany R, Juhasz T. Disturbed spermatogenic signaling in pituitary adenylate cyclase activating polypeptide-deficient mice. Reproduction. 2018 Feb;155(2):129-139. 

Tanimoto K1, Suzuki AOhno SHonda KTanaka NDoi TNakahara-Ohno MYoneno KNakatani YUeki MYanagida TKitamura RTanne K. Hyaluronidase expression in cultured growth plate chondrocytes during differentiation. Cell Tissue Res. 2004 Nov;318(2):335-42.

Deschrevel B1, Lenormand HTranchepain FLevasseur NAstériou TVincent JC Hyaluronidase activity is modulated by complexing with various polyelectrolytes including hyaluronan. Matrix Biol. 2008 Apr;27(3):242-53.

Analysis of the expression levels of the chondrogenic markers for confirmation of the differentiation stages (e.g. type II collagen, Indian hedgehog and type X collagen) and of the enzymes forming the ECM would verify the conclusion of the study.

We have already published some results to show the time dependency of aggrecan and collagen type II (Zakany et al., 2005) and the presence of collagen type X in 3 day old chondrogenic cultures is also proven (Juhasz et al., 2015). We have also shown that PACAP addition has a positive effect on the protein expression on collagen type II, aggrecan and Sox9 but it prevents elevation of collagen type X expression (Juhasz et al., 2015). Our previous experiments indicate that the decreased expression of these proteins in mechanical stress (Juhasz et al., 2015) and in oxidative stress generated with H2O2 (Juhasz et al., 2014) can be prevented by PACAP via normalizing their protein expression. Indeed, there was a missing link in balancing processes, therefore, we hypothesized that PACAP can be beneficial in circumstances where matrix degradation occurs in articular cartilage (i.e. during inflammation or in case of severe cellular stress) and we aimed to prove this idea in the current experiments. Analysis of the changes of chondrogenic marker gene expression would not carry novel finding as we have published such data in the referred papers.

Zákány R, Szíjgyártó Z, Matta C, Juhász T, Csortos C, Szucs K, Czifra G, Bíró T, Módis L, Gergely P. Hydrogen peroxide inhibits formation of cartilage in chicken micromass cultures and decreases the activity of calcineurin: implication of ERK1/2 and Sox9 pathways. Exp Cell Res. 2005 Apr 15;305(1):190-9.

Juhász T, Szentléleky E, Somogyi CS, Takács R, Dobrosi N, Engler M, Tamás A, Reglődi D, Zákány R. Pituitary Adenylate Cyclase Activating Polypeptide (PACAP) Pathway Is Induced by Mechanical Load and Reduces the Activity of Hedgehog Signaling in Chondrogenic Micromass Cell Cultures. Int J Mol Sci. 2015 Jul 29;16(8):17344-67.

Juhász T, Matta C, Katona É, Somogyi C, Takács R, Gergely P, Csernoch L, Panyi G, Tóth G, Reglődi D, Tamás A, Zákány R. Pituitary adenylate cyclase activating polypeptide (PACAP) signalling exerts chondrogenesis promoting and protecting effects: implication of calcineurin as a downstream target. PLoS One. 2014 Mar 18;9(3):e91541.